# Loss of SRSF2 triggers hepatic progenitor cell activation and tumor development in mice

Chang Zhang[1,5], Lei Shen [2,5], Wei Yuan[3], Yuguo Liu[1], Ruochen Guo[1], Yangjun Luo[1], Zheng Zhan[1], Zhiqin Xie[1], Guohao Wu[2], Wenwu Wu[4 ✉] & Ying Feng[1,4 ✉]

Splicing factor SRSF2 is frequently mutated or up-regulated in human cancers. Here, we observe that hepatocyte-specific deletion of *Srsf2* trigger development of hepatocellular carcinoma (HCC) in mice, which also involves inflammation and fibrosis. Importantly, we find that, when compensatory hepatocyte proliferation is impaired, activation of hepatic progenitor cells (HPCs) play an important role in liver regeneration and tumor formation. Moreover, the cells of HCC- bearing livers display both HPC and hepatocyte markers, with gene expression profiling suggesting HPC origin and embryonic origin. Mechanically, we demonstrate that levels of oncofetal genes insulin-like growth factor 2 (Igf2) and H19 are significantly increased in the tumors, likely due to decreased DNA methylation of the *Igf2/ H19* locus. Consequently, signaling via the Igf2 pathway is highly activated in the tumors. Thus, our data demonstrate that loss of Srsf2 triggers HPC-mediated regeneration and activation of oncofetal genes, which altogether promote HCC development and progression in mice.

[1] CAS Key Laboratory of Nutrition, Metabolism and Food Safety, Shanghai Institute of Nutrition and Health, Shanghai Institutes for Biological Sciences, University of Chinese Academy of Sciences, Chinese Academy of Sciences, Shanghai, China. [2] Department of General Surgery, Zhongshan Hospital, Fudan University, Shanghai, China. [3] Department of Pathology, Zhongshan Hospital, Fudan University, Shanghai, China. [4] State Key Laboratory of Subtropical Silviculture, Zhejiang A&F University, Lin'an, Hangzhou, China. [5] These authors contributed equally: Chang Zhang, Lei Shen. ✉email: wwwu@zafu.edu.cn; fengying@sibs.ac.cn

The liver is a central organ for homeostasis and performs a variety of functions. Most of the functions are carried out by the liver parenchymal cells (hepatocytes), which account for approximately 80% of the total liver volume. Moreover, quiescent hepatocytes can re-enter cell cycle and proliferate to maintain the organ's functional integrity when the liver suffers from acute injuries, a process referred to as compensatory proliferation[1]. The main non-parenchymal cells of the liver include oval cells, biliary epithelial cells (BECs), hepatic stellate cells and kupffer cells[2,3]. Oval cells are commonly referred to as HPCs in rodents, as they possess bipotential differentiation capacity toward both hepatocytes and BECs. They normally reside in biliary ducts and can be activated by impairment of hepatocyte replicative potential during chronic liver damage, which is the so-called oval response[1,2]. However, recent studies demonstrate that the resident BECs can also be activated and differentiate to hepatocytes under injury conditions that closely mimic human chronic liver diseases[4–6]. Thus from a broad perspective, both oval cells and BECs are considered to be HPCs, based on their histological characteristics of BEC marker positivity and ectopic emergence in the parenchymal region[1].

A broad range of hepatic insults such as hepatitis virus, alcohol, fat accumulation, and drug toxicity can damage the liver to trigger steatohepatitis, fibrosis, cirrhosis, and even development of liver tumors. Primary malignant liver tumors include HCC and cholangiocarcinoma, which accounts for approximately 90% or 10–15%, respectively[7]. HCC is an aggressive primary liver cancer that has been recognized as a leading cause of death among patients with cirrhosis[8]. Thorgeirsson and colleagues found that a distinct subtype of aggressive HCC could express markers of HPCs, suggesting that HCC of this subtype might arise from HPCs[9]. Consistent with this, hepatic deletion of PR-SET7 caused HPC-mediated regeneration, which could contribute to spontaneous HCC on the mice[10]. However, lineage tracing experiments suggested that HCC originated from hepatocytes but not the biliary compartment[11]. On the other hand, using human data and various hepatocarcinogenesis mouse models, Tummala and his colleges have showed that HCC progression was often accompanied by HPC activation[12]. Thus, different cell of origin might reflect various carcinogenesis during HCC development.

SRSF2 has been reported to be involved in a multiple biological processes and tumor progression. Inactivation of Srsf2 could trigger defects in the development of thymus and heart[13,14]. Mutations at proline 95 substituted with histidine (Srsf2$^{P95H}$) were sufficient to induce in myelodysplastic syndromes (MDS) in inducible Mx1-Cre Srsf2$^{P95H/WT}$ knock-in mice[15,16]. This was consistent with the fact that SRSF2 mutations played a major role in the pathogenesis of MDS[17]. Moreover, high levels of SRSF2 contributed to HCC progression and were associated with poor prognosis in patients[18].

We previously described the generation of conditional Srsf2 knockout mice (HKO)[19] by crossing mice carrying loxP-flanked Srsf2 alleles (Srsf2$^{f/f}$) with Alb-cre trans-genetic mice, which mediate efficient Cre recombination in hepatocytes. In this study, we focused on more in-depth analysis of the function of Srsf2 in adult livers. Here we observed that, when the compensatory proliferation of hepatocytes was impaired during chronic liver injury, HPCs were activated and expanded. HKO mice at 12 months of age and over developed HCC, which displayed characteristics of both HPC and hepatocyte markers. RNA-seq analysis further revealed that tumors from Srsf2 HKO mice expressed both HPC markers and oncofetal markers, which was similar to what observed in human aggressive HCC. Significantly, Igf2 and H19, two oncofetal and imprinting genes, were highly expressed in the tumors, accompanied by activation of PI3K/Akt and MAPK/Erk signaling pathways. Accordingly, methylation at the fetal promoter of Igf2 and imprinting control region (ICR) of H19 was reduced, which accounts for the high levels of Igf2 or H19, respectively. Taken together, we have concluded that deletion of Srsf2 triggers HPC proliferation, activation of oncofetal genes and multiple signaling pathways, finally leading to HCC development in mice.

## Results

**Srsf2 deletion resulted in HCC development in aged mice.** Ablation of Srsf2 led to severe liver injury so that the majority of HKO mice died 3 weeks after birth[19]. In order to rule out the possibility that mutant offspring died from insufficient competition for milk, we decided to take most of control littermates out to other cages shortly after birth, which would allow the mutant offspring well taken care of. Under the meticulous care, among a total of 136 HKO mice, 40 mice survived to adulthood (Supplementary Data 1, Fig. 1a). The surviving mice gained weight in a delayed manner until 3 months of age when compared with controls (Supplementary Data 1, Fig. 1b). The liver weight/body weight ratio and spleen mass/body mass ratio were significantly higher in the HKO mice than age-matched controls (Supplementary Data 1, Fig. 1c, d). High concentration of ALT and AST were also observed in HKO mice compared with control animals (Supplementary Fig. 1), indicating that chronic liver injury persists in HKO mice.

Dissection of livers from 12-month-old HKO mice surprisingly revealed the presence of numerous visible cancerous nodules of different sizes (Fig. 1e). No tumors were observed in the liver of Srsf2$^{f/f}$ mice, which were used as wild-type controls. The penetrance of liver tumors was observed in 18 from a total of 25 mice at ages 12M and over. And smaller hyperplastic nodules were even observed in 6M HKO mice. HE staining revealed that normal hepatic cell structure disappeared in the tumor section of 12M HKO mice, instead multiple nodular lesions with unregular structures appeared (black dotted circles) (Fig. 1f). Cells within the lesion were arranged in beams and cords (Fig. 1g), and they demonstrated increased cellularity with marked hyperchromatism when compared with cells in the non-lesioned tissue (Fig. 1h). All these indicated that lesions in the HKO livers were morphologically HCC.

Consistent with the HE staining, immunostaining further showed that Ki67-postive cells were observed within the tumor, much less observed in the tissue adjacent to the tumor, but not observed in the control liver sections (Supplementary Data 2, Fig. 1i). Alpha fetal protein (Afp) and glutamine synthetase (Gs), two molecular markers of HCC, were observed in the tumor sections (Fig. 1i). P62, a ubiquitin-binding autophagy receptor, which was required for HCC induction in mice[12], was highly expressed in the tumor (Fig. 1i). In addition, Sirius-Red staining and CD45 staining revealed that fibrosis and inflammation occurred in non-tumorous sections but not in tumorous sections (Fig. 1j). Taken together, loss of Srsf2 resulted in chronic liver injury, fibrosis and HCC development in mice.

**Impaired hepatocyte proliferation occurred in HKO mice.** Human HCC often arises from chronic inflammation[20]. To explore the underlining mechanisms for Srsf2 deficiency-induced HCC, we examined livers from mutant mice between 1 and 3 months of age. As shown in Fig. 2a, altered architectures, fibrosis and inflammation were all observed in the mutant mice, compared with control animals. Because of the essential role for Srsf2 in maintaining genomic stability during mammalian organogenesis[21], we next wanted to examine whether double-stranded DNA breaks (DSBs) occur in Srsf2 HKO livers. Indeed, phosphorylated γH2AX, an indicator usually used in monitoring

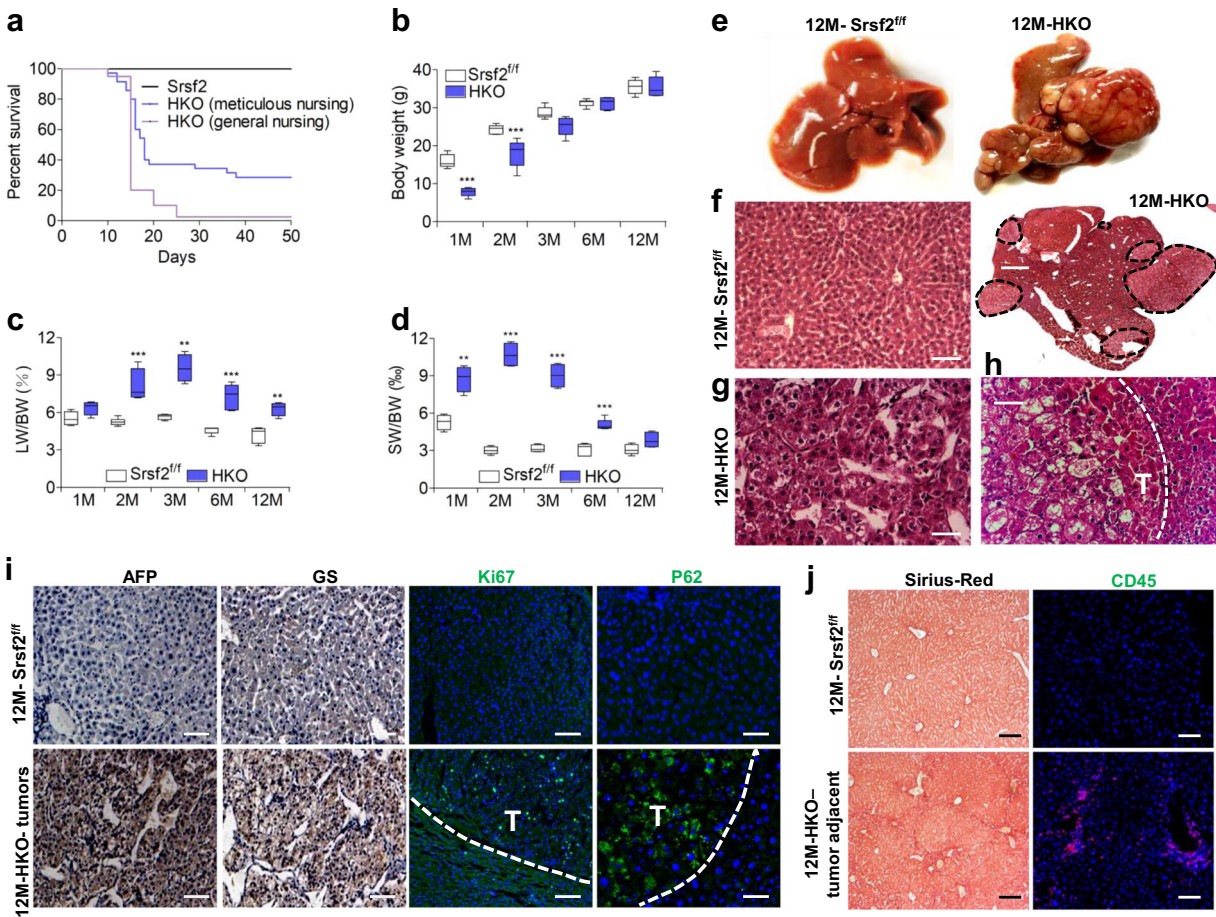

**Fig. 1 Srsf2 deletion resulted in HCC development in the aged HKO mice. a** Survival curve analysis of Srsf2^f/f and HKO mice under the general nursing or meticulous nursing. **b** Comparison of body weight between Srsf2^f/f and HKO mice at indicated months (M) of age (*n* = 5). **c, d** Comparison of ratios of liver weight (LW) or spleen weight (SW) to body weight (BW) between Srsf2^f/f and HKO mice at indicated ages (*n* = 4). **e** Representative livers of Srsf2^f/f and HKO mice at 12M of age. **f** Representative hematoxylin/eosin (HE)-stained liver sections from Srsf2^f/f (scale bars, 50 μm) and HKO mice with tumors (scale bars, 500 μm) at 12M (*n* = 3). **g, h** Enlarged HE staining of HKO mice with tumors shown in (**f**) (scale bars, 50 μm). **i** Representative immunostaining of Afp, Gs, Ki67, or P62 of liver sections (scale bars, 50 μm) from 12M Srsf2^f/f and tumors of 12M mutant mice (*n* = 3). **j** Sirius-Red staining (scale bars, 100 μm) and CD45 staining (scale bars, 50 μm) of Srsf2^f/f livers and non-tumorous areas of 12M mutant mice (*n* = 3). *P < 0.05, **P < 0.01, and ***P < 0.001 vs. matched control by Student's *t* test (**b, c, d**).

DSBs, was obviously observed in livers from HKO mice at 1 month of age. Consistent with the presence of DSBs, increased apoptosis, cell cycle arrest and autophagy were all evident in the mutant livers compared with control animals, as addressed by TUNEL assay, P21 and P62 staining, respectively (Supplementary Data 2, Fig. 2b). Interestingly, positive signals for these markers gradually declined in livers from 2M HKO mice until totally diminished in livers from mice at 3 months of age (Fig. 2b).

Next we wanted to ask whether mutant hepatocytes would enter the cell cycle to facilitate liver repair. To this end, we examined expression of the proliferation marker Ki67 in livers from mice between 1 and 3 months of age. As shown in Fig. 2c, weak proliferation signals were still detected in 1M livers from control mice, but no signals were present in 2M or 3M livers. In contrast, mutant livers underwent remarkable proliferation at 1 month old, which gradually declined at 2 months and almost diminished at 3 months of age. These findings indicated that severe injury triggered hepatic compensatory proliferation in the mutant livers.

Next we carried out double-immunostaining with Ki67 and the hepatic marker Hnf4α or Ki67 and CD45, using 1M liver sections. Although majority of proliferative cells belonged to hepatocytes, CD45^+ inflammatory cells were also observed to proliferate in the

mutant livers (Fig. 2d). In order to provide more evidence that there was an increased proliferation of hepatocytes in the HKO mice, we injected HKO mice at 1 month of age with EdU, and analyzed liver sections using EdU assay after 5 days of injection. Fully in line with the above results, more Edu^+/Hnf4a^+ cells were observed in the liver section of HKO mice than in controls (Fig. 2e, top arrows), and Edu^+/DAPI^+ cells lacing Hnf4a signals also exist in the HKO livers (Fig. 2e, bottom arrows). Meanwhile, double-immunostaining with P21 and Hnf4α manifested that hepatocytes in the HKO livers also suffered from cell cycle arrest, as P21 signals are observed in hepatocytes, but not colocalized with Ki67 signals (Fig. 2f). These findings indicated that the proliferative capacity of hepatocytes in the mutant livers was compromised.

**Adult HPCs were activated and expanded in the mutant livers.** As inflammation and fibrosis persisted, and hepatocyte-mediated regeneration process was compromised, next we wanted to examine whether the HPCs could be activated in the mutant livers. To this end, we performed immunostaining with antibodies against HPC maker A6. This revealed massive expansion and migration of adult HPCs in the 2M mutant livers, which

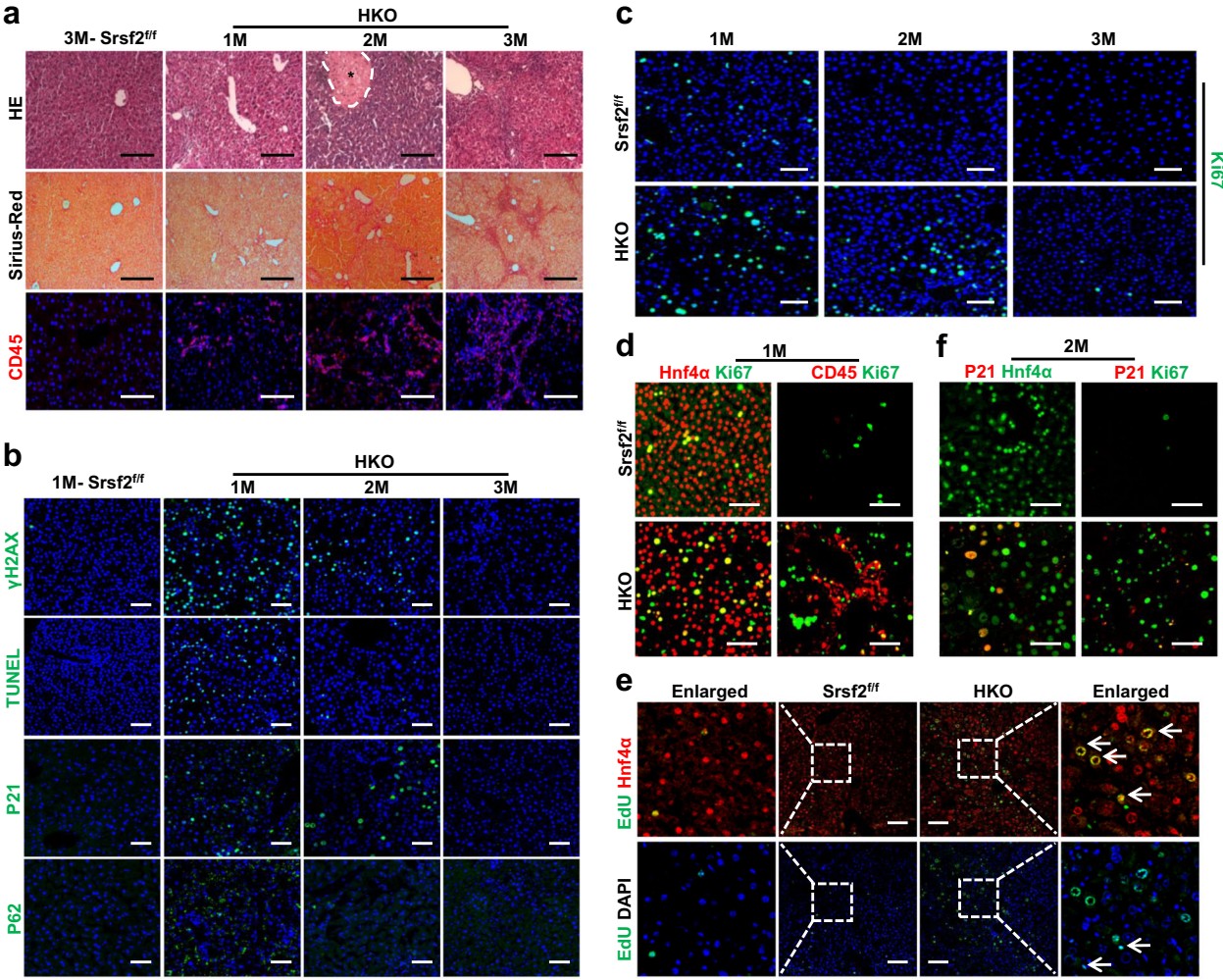

**Fig. 2 Impaired hepatic proliferation was observed in the mutant livers. a** Representative HE (scale bars, 100 μm), Sirius-Red (scale bars, 200 μm) or CD45 (scale bars, 100 μm) staining of livers from 1-3M mutant mice and 3M Srsf2$^{f/f}$ mice ($n = 3$). Asterisk indicated necrotic area in the 2M HKO liver. **b** Representative staining of γH2AX, TUNEL, P21 or P62 of livers from 1M-3M mutant mice or 1M Srsf2$^{f/f}$ mice (scale bars, 50 μm) ($n = 3$). **c** Representative Ki67 immunostaining of livers from 1-3M Srsf2$^{f/f}$ mice or mutant mice (scale bars, 50 μm) ($n = 3$). **d** Representative double-immunostaining with Hnf4α/Ki67 or CD45/Ki67 for 1M mutant and control livers (scale bars, 50 μm) ($n = 3$). **e** EdU assay. EdU in the liver sections were detected by labeling with Alexa-488 (Green) and nuclei were stained with Hnf4a (red) or DAPI (blue). Representative merged images were shown with Edu/Hnf4a or Edu/DAPI (scale bars, 100 μm) ($n = 3$). **f** Representative double-immunostaining with P21/Hnf4α or P21/Ki67 for 2M mutant and control livers (scale bars, 50 μm) ($n = 3$).

looked like a fan infiltrating into the parenchymal tissue, while only a few signals were present around the portal vein in the control livers (Fig. 3a). The activation and migration of HPCs were further confirmed by staining with other HPC markers CK19 or Sox9. Double staining with A6 and CK19, A6 and Sox9, or CK19 and Sox9 antibodies revealed that vast majority of cells stained positively with either A6 or Sox9 or CK19 or combinations of them (Fig. 3a). Fully in line with these results, qPCR analysis demonstrated a dramatic increase in mRNA levels of genes that were highly expressed in HPCs, such as Sox9, CK19, CD44, Proml, CK7, or Spp1 in 2M and 3M HKO livers compared with controls (Supplementary Data 2, 3, Fig. 3b).

Staining of Ki67 and CK19 showed that adult HPCs have strong proliferative capacity in the 2M mutant livers compared with control livers (Fig. 3c). It was known that activation of mesenchymal or inflammatory cells surround HPCs, or proteins of the extracellular matrix are important for HPC expansion[22]. Indeed, we found that stellate cells, which can secret αSMA, and CD45$^{+}$ inflammatory cells were tightly associated with CK19$^{+}$ cells, as well as excessive amounts of laminin proteins in the

vicinity of the A6$^{+}$ cells (Fig. 3c). In addition, the growth factor Fgf7 has been observed to be required for stimulation of HPCs[23] and a large quantity of Fgf7 protein was also observed in the vicinity of the A6$^{+}$ cells (Fig. 3c). Accordingly, mRNA levels of Fgf7 and Hgf, another factor implicated in the regulation of HPC proliferation[24], were elevated in 2M and 3M HKO livers compared with controls (Supplementary Data 2, 3, Fig. 3d). Taken together, these data strongly indicated that adult HPCs were activated and expanded in the mutant livers compared with controls.

**Srsf2 deletion in the adult liver resulted in HPC activation.** Next we wanted to ask whether inactivation of Srsf2 in adult livers directly triggers HPC activation and proliferation. To this end, we crossed Srsf2$^{f/f}$ mice with Mx1-cre mice to obtain Mx1cre-Srsf2$^{f/f}$ mice, which could be induced by intraperitoneal injection with polyI:C to produce Srsf2$^{-/-}$ mice with conditional deletion of *Srsf2* (Fig. 4a). As DDC treatment could lead to HPC proliferation, we also fed Srsf2$^{f/f}$ mice with DDC diet for 6 weeks

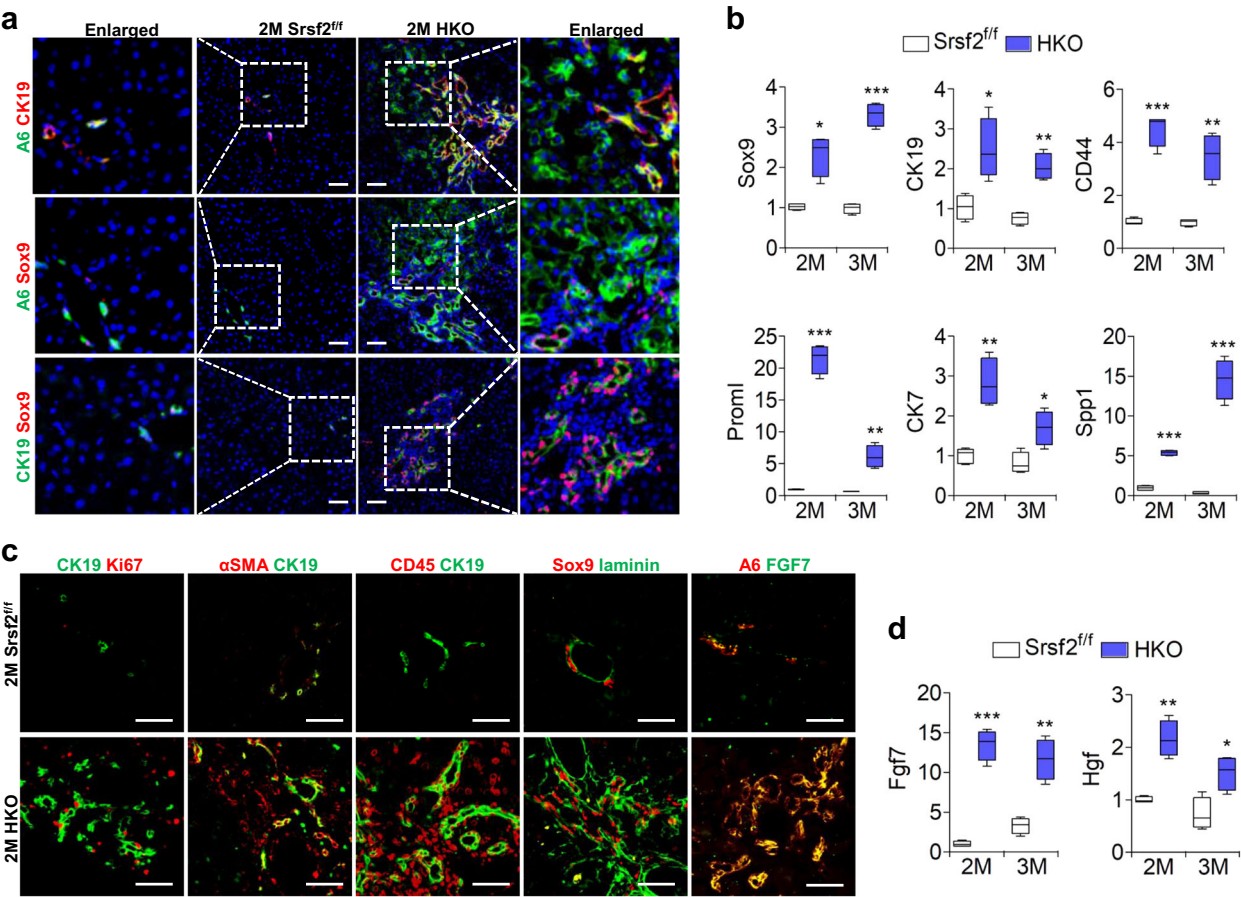

**Fig. 3 Chronic liver injury resulted in HPC activation and proliferation. a** Representative double-immunostaining with A6/CK19, A6/sox9 or CK19/sox9 for mutant and control livers from mice at 2M (scale bars, 50 µm) ($n = 3$). **b** Comparison of mRNA levels of HPC marker genes between mutant and control livers of 2M or 3M ($n = 4$). **c** Representative double-immunostaining with CK19/Ki67, αSMA/CK19, CD45/CK19, sox9/laminin or A6/FGF7 for mutant and control livers of 2M (scale bars, 50 µm) ($n = 3$). **d** mRNA levels of Fgf7 and Hgf measured by qPCR at 2M or 3M ($n = 4$). *$P < 0.05$, **$P < 0.01$, and ***$P < 0.001$ vs. matched control by Student's $t$ test (**b**, **d**).

as positive controls (Fig. 4a). At 8 weeks after polyI:C injection, all the mice were sacrificed for analysis. Western blotting analysis demonstrated that Srsf2 protein was efficiently deleted from the livers from Srsf2$^{-/-}$ mice at 4 weeks or 8 weeks after poly I:C injection (Fig. 4b, Supplementary Fig. 2). Significantly, altered liver architectures and fibrosis were evident in both Srsf2$^{-/-}$ and DDC-treated mice compared with controls (Fig. 4c). Increased levels of DSB, apoptosis, autophagy and cell cycle arrest were observed in Srsf2$^{-/-}$ mice when compared with DDC-treated mice (Fig. 4d, e). Moreover, significantly elevated amounts of A6$^+$/CK19$^+$ cells or increased Ki67 signals in both HPCs and hepatocytes were also detected in these mice compared with controls and DDC-treated mice (Supplementary Data 2, 3, Fig. 4d, e). Taken together, these findings have provided strong evidence that impaired Srsf2 expression in adult livers could directly result in HPC activation and proliferation, the effects apparently stronger than DDC treatment.

**A6$^+$/Hnf4α$^+$ bi-phenotypic cells were observed in the mutant.** As long-term DDC treatment has been observed to promote differentiation of BECs into hepatocytes[4,5], we wondered whether HPCs could be induced to express markers of hepatocytes in the HKO mice. To this end, we first exposed the Srsf2$^{f/f}$ mice to DDC-containing diet for 4.5 months. We found that A6$^+$ cells could express the hepatic marker Hnf4α, and notably some of double-positive cells were binucleated (yellow arrow), a hallmark

of mature hepatocytes (Fig. 5a). Next, we immunostained liver sections from 1M, 2M or 3M mutant mice with antibodies against A6 and Hnf4α. While there were no A6 positive signals in 1M livers, cells around the portal vein (PV) were stained positive for A6, but negative for Hnf4α in the 2M mutant livers (white arrowheads). And cells in the parenchymal section adjacent to PV were stained double positive for A6$^+$ and Hnf4α$^+$ in 2M or 3M mutant livers (white arrows). Of even greater significance, A6$^+$/ Hnf4α$^+$ binuclear cells were observed in the 3M mutant livers (yellow arrow) (Fig. 5a). In addition, the periodic acid–Schiff (PAS) staining showed that the area stretched from PV was glycogen-staining positive in the 2M HKO livers compared with controls (Fig. 5b). This likely reflected that those cells had features of metabolically functional hepatocytes. As macrophages play a critical role in HPC specification[25], co-staining with macrophage marker F4/80 and HPC marker Sox9 or CK19 revealed that macrophages were substantially existed closely to HPCs in the HKO livers (Fig. 5c). These results strongly indicated that A6$^+$/ Hnf4α$^+$ bi-phenotypic cells were present beyond periportal regions in the mutant livers.

**Tumor cells displayed both HPC and hepatocyte markers.** Next we wanted to examine levels of both HPC markers and hepatocyte markers expressed in tumors developed in 6M or 12M mutant mice. Double immunostaining revealed that A6$^+$/Hnf4α$^+$ cells were present within the tumor section, but not observed

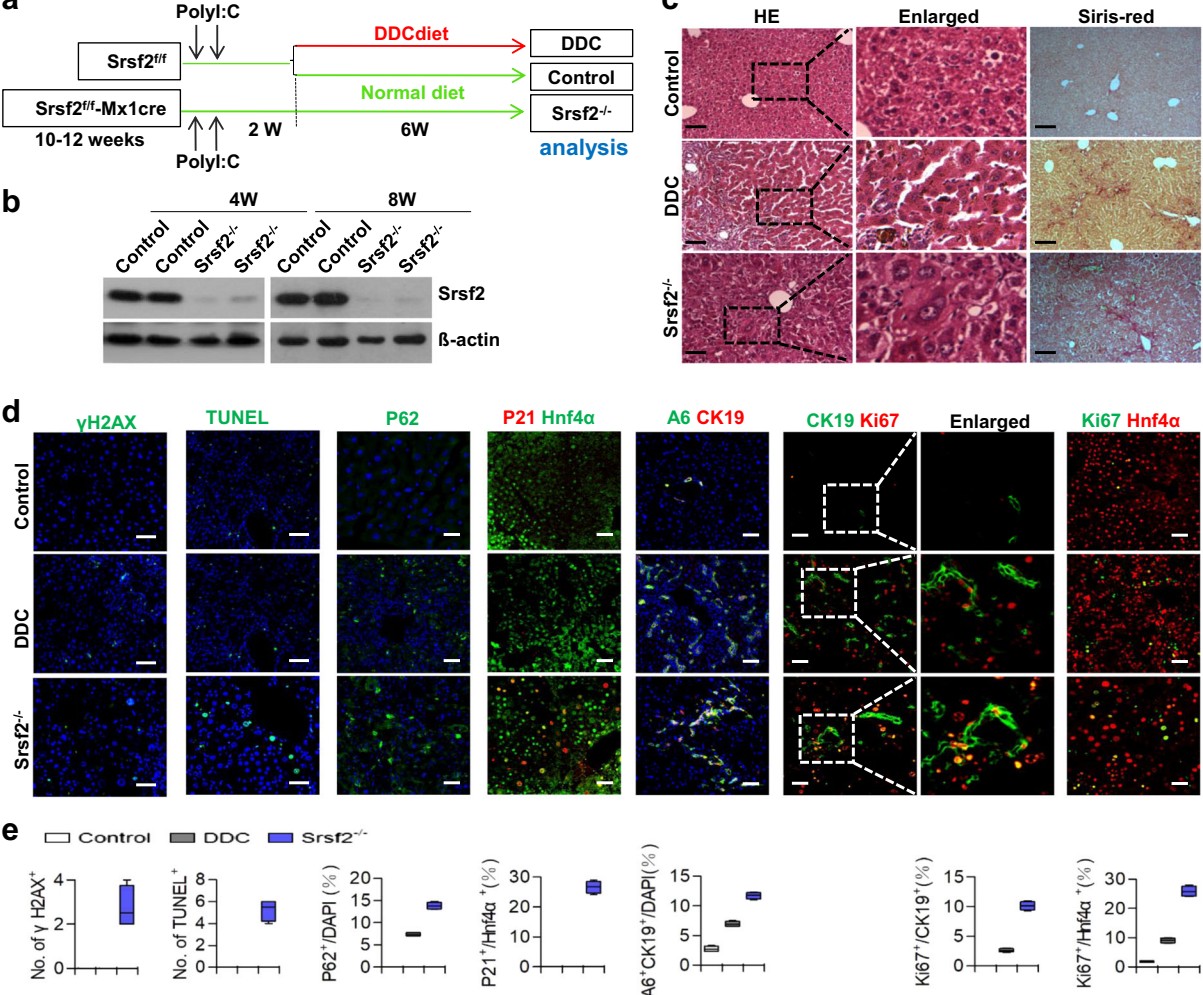

**Fig. 4 Inactivation of Srsf2 in the adult liver activated HPC proliferation. a** Schematic design for production of Srsf2$^{-/-}$ mice, DDC-treated mice or control mice after injection of polyI:C and fed with DDC diet or chow diet. **b** Detection of Srsf2 protein levels at 4- and 8-week (W) after polyI:C injection by western blotting analysis. The original gel image was shown in Supplementary Fig. 2. **c** Representative HE (scale bars, 50 μm) and Sirius-Red (scale bars, 100 μm) staining of livers from control, DDC-treated or Srsf2$^{-/-}$ mice after injection with polyI:C for 8 weeks. **d** Representative staining of γH2AX, TUNEL or P62 and double-immunostaining with P21/Hnf4α, A6/CK19, CK19/Ki67 or Ki67/Hnf4α of livers at 8 weeks after polyI:C injection (scale bars, 50 μm) ($n = 3$). **e** Images shown in (**d**) were qualified at the bottom.

either in tissues adjacent to tumors or in control livers (Supplementary Data 2, Fig. 6a). Moreover, co-staining with another HPC marker Sox9 showed that those Hnf4α$^+$ cells or A6$^+$ cells in the tumors all expressed Sox9 proteins (Fig. 6a). However, co-staining with CK19 and Hnf4α showed that CK19 proteins were not expressed in Hnf4α$^+$ cells within the tumors (Fig. 6a). Taken together, these findings have provided strong evidence that tumors developed in the mutant mice had characteristics of both HPCs and hepatocytes. Importantly, although *Srsf2* gene was also deleted in the adult livers as well as in the livers of HKO mice at 2 weeks of age (Supplementary Fig. 3), recurrence of Srsf2 proteins were detected in the livers at 2 months of age and over, compared to complete lack of Srsf2 proteins in 2W and 1M livers, as addressed by western blotting analysis (Fig. 6b, Supplementary Fig. 4). Consistent with re-expression of Srsf2 proteins, altered alternative splicing events, which were detected in the 2 W HKO livers[19], were recovered to their wild-type patterns in livers of HKO mice at 2M, 6M or 12M (Supplementary Fig. 5). In addition, no compensatory changes were observed for other members of SR family (Supplementary Fig. 6).

As the gene expression profiling could provide insight into the mechanism underlining the tumors, next we decided to take advantage of RNA-seq platform to determine differentially expressed genes in tumors from 12M mutant mice, compared with non- tumor liver tissues or control liver tissues at the same age (Supplementary Data 4). A large scale of genes was observed to be up-regulated and down-regulated in the tumors compared with non-tumor tissues (Fig. 6c). Up-regulated genes include progenitor markers (*Sox4*, *Sox9*, *Itga6* or *Spp1*); oncofetal markers (*Gas5*, *Igfbp1*, *Afp*, *Igf2* or *H19*), and HCC progression markers (*Plau*, *Golm1* or *Vim*). Down-regulated genes contain markers enriched in mature hepatocytes (*Tat*, *Cyp7a1*, *Apoa5*, *Pck1*, *Cyp3a11*, *Mup3* or *Acsl1*) and tumor suppressors (*Lect2*, *Wfdc17* or *Efemp1*) (Fig. 6d).

Next we obtained differentially expressed genes between 2M HKO livers and age-matched control livers by RNA-seq analysis (Supplementary Data 4). We also analyzed public microarray data for differentially expressed genes between adult liver progenitor cells (LPCs) and control cells (Supplementary Data 4)[26]. After comparison of these three datasets, we found that 20% up-regulated

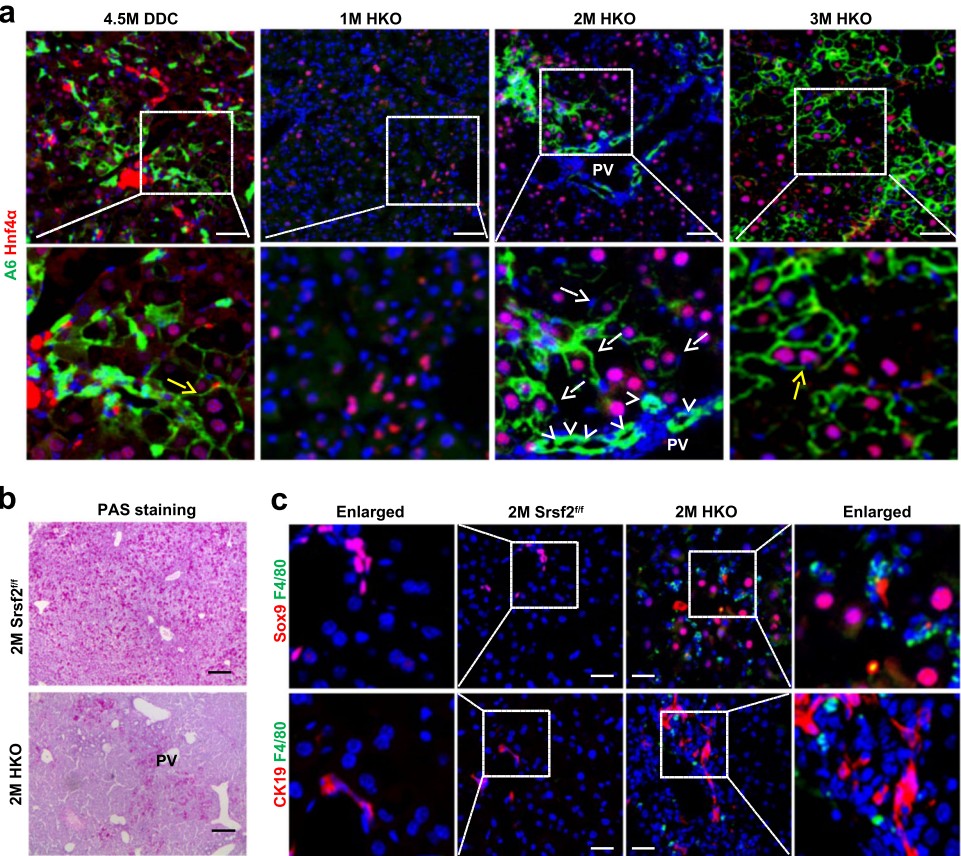

**Fig. 5 A6+/Hnf4α+ bi-phenotypic cells were observed in the mutant livers. a** Representative double-immunostaining with A6/Hnf4α of livers from Srsf2f/f mice treated with DDC for 4.5M or from HKO mice at 1M-3M (scale bars, 50 μm) (*n* = 3). **b** Representative PAS staining of 2M control and mutant livers (scale bars, 200 μm) (*n* = 3). **c** Representative double-immunostaining with Sox9 or CK19 and F4/80 of 2M mutant and control livers (scale bars, 50 μm) (*n* = 3).

genes in the 12M tumors was observed to be expressed in the LPCs, and 34.6% in the 2M HKO livers (Fig. 6e). Significantly, overlapped genes among the three datasets also included progenitor markers, hepatic oncofetal markers and HCC progression markers. In the Venn diagram of down-regulated genes, the 12M tumors displayed a much higher overlap ratio of 44.5% to the LPCs, or 42.5% to the 2M HKO livers (Fig. 6f). Expectedly, those markers for mature hepatocytes or for tumor suppressors were identified in the overlaps. The aggressive human HCC, which was associated with poor prognosis, also expressed oncofetal genes such as *IGF2*, *AFP*, *H19* or *IGFBF1* and HPC makers[9]. Taken together, all these results provided strong evidence that HCC developed on the Srsf2 HKO mice contains both HPC origin and embryonic origin.

**Highly activated Igf2 signaling was observed in the tumors.** We next wanted to uncover the underlying mechanisms through which *Srsf2* deletion promoted HCC development in the context of chronic liver injury. Dramatically, two oncofetal genes Igf2 and H19 were on the top of the most strongly up-regulated transcripts based on RNA-seq analysis of 12M tumors versus non-tumors (Table 1). Afp was also presented among these ten genes. qPCR confirmed that the relative cycles of Igf2 and H19 significantly decreased in the tumor compared with controls and non-tumor livers (Supplementary Data 2, 5, Fig. 7a).

To explore whether high expression of Igf2 and dH19 was directly related with loss of Srsf2 proteins in the livers, we chose 3 time points to analyze their expression levels. As shown in Supplementary Fig. 7, levels of Igf2 and H19 were significantly

decreased in the 2W HKO livers compared to controls when knockout livers went through massive cell death due to loss of Srsf2 proteins[19]. However, levels of Igf2 or H19 were observed to be up-regulated more than 20-fold or 10-fold, respectively, in the 1M HKO livers when they displayed compensatory hepatocyte proliferation but still no expression of Srsf2 proteins (also see Fig. 2c). Moreover, mRNA levels increased hundreds of times for Igf2, or thousands of times for H19 in the 2M HKO livers when they displayed HPC proliferation and recurrence of Srsf2 proteins (also see Fig. 3a). Thus, up-regulation of Igf2 and H19 was not directly connected with Srsf2 loss, instead closely related with liver regeneration in the Srsf2 knockout mice model, and this phenomena was consistent with previous literatures[10,27].

IGF2 is an effective predictor of cancer risks as its over-expression occurred in many cancers and was associated with a poor prognosis[28]. IGF1 receptor (IGF1R) was thought to play a predominant role in the IGF2 signaling, which led to PI3K/AKT or MAPK/ERK pathway activation[28]. Gene Ontology and KEGG analysis revealed that both MAPK cascade and PI3K/Akt signaling pathway were activated in the malignant tumors (Fig. 7b). We next wanted to examine the protein levels of Igf2 and its relevant pathway changes in the tumors (Supplementary Data 2, Supplementary Fig. 8). As shown in Fig. 7c, Igf2 was significantly up-regulated in the 12M tumors but undetectable in non-tumors and control livers. Consistent with high levels of Igf2, phosphorylated Igf1r proteins were also highly expressed in the tumors, indicating that Igf2/Igf1r signaling was activated.

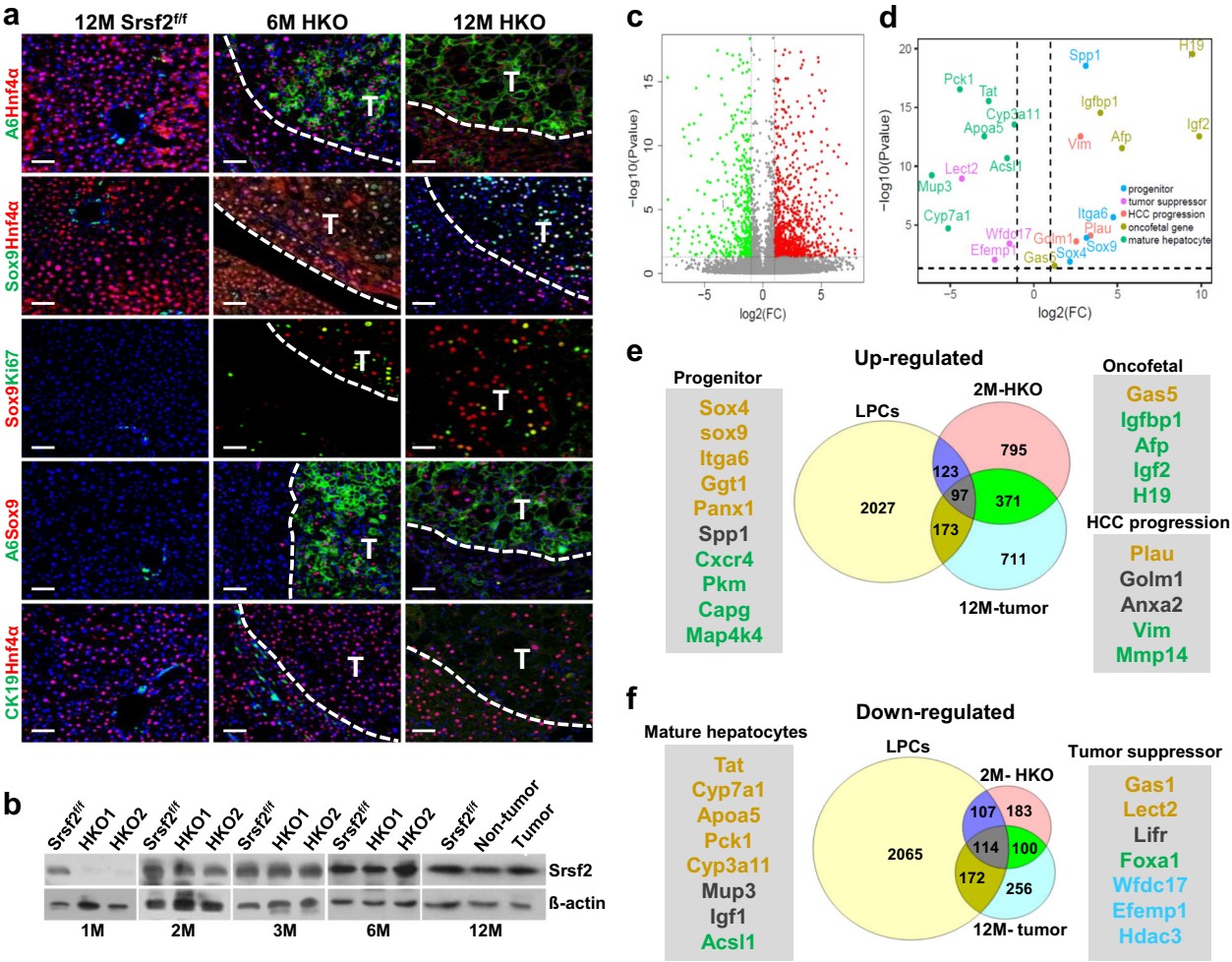

**Fig. 6 Malignant tumors displayed both HPC and hepatocyte markers. a** Representative double-immunostaining with A6/Hnf4α, Sox9/Hnf4α, A6/Sox9, CK19/Hnf4α or Sox9/Ki67 in 6M, 12M HKO livers with tumors or 12M control livers (scale bars, 50 µm) (n = 3). **b** Srsf2 protein levels were detected by western blotting analysis at indicated ages of control and mutant livers. **c** Volcano plot showed up-regulated (red) and down-regulated (green) genes of tumors versus non-tumors of 12M mutant mice. Significance cutoff value was set at P = 0.05 in −log10 scale (horizontal dashed line) and fold change cutoffs at −1 and 1 in log2 scale (vertical dashed lines). **d** Volcano plot showed dots corresponding to up-regulated genes including progenitor markers (light-blue), oncofetal gene markers (brown) and HCC progression markers (pink), and down-regulated genes containing mature hepatocyte markers (green) and HCC suppressor markers (purple). **e, f** Comparison of differentially expressed genes among 12M tumors, 2M HKO livers and adult liver progenitor cells (LPCs).

**Table 1 List of fold changes of the top 10 up-regulated genes revealed by RNA-seq analysis: tumor(T) vs non-tumor (N).**

| Gene Name | N (FPKM) | T (FPKM) | T/N (ratio) |
|---|---|---|---|
| Igf2 | 0.59 | 534.40 | 905.76 |
| H19 | 12.06 | 8233.70 | 682.73 |
| Csn3 | 1.10 | 86.90 | 79.00 |
| Cbr3 | 2.72 | 111.10 | 40.85 |
| Mogat2 | 1.40 | 56.50 | 40.36 |
| Cxcr4 | 2.44 | 95.70 | 39.22 |
| Afp | 1.45 | 53.50 | 36.90 |
| Gpx2 | 1.75 | 56.50 | 32.29 |
| Ly6c1 | 2.03 | 58.00 | 28.57 |
| Serpine1 | 2.53 | 67.80 | 26.80 |

Meanwhile, phosphorylated proteins of Akt, Erk, Jnk or Stat3 were all increased in the tumors compared with non-tumors and controls. High levels of Jun proteins were also observed in the tumors. In summary, Igf2/Igf1r signaling and its subsequent activation of multiple signaling pathways facilitated tumorigenesis in the HKO livers.

The Igf2 was clustered with H19 on the chromosome 7 in mice controlled by an ICR located in the 5′-flank of H19 (Supplementary Fig. 9), which is a differentially methylated region[29,30]. As expected, the methylation level decreased nearly 30% in 12M tumors than non-tumors (Supplementary Data 1, Fig. 7d), which correlates with the elevated H19 levels. As IGF2 transcripts initiated from the fetal promoter were observed in human cancer cells[31], the predicted CpG islands are also present in this promoter region. Fully in line with high levels of Igf2 in tumors, the methylation of the fetal promoter designated as P2, which harbors predicted CpG islands (Supplementary Fig. 6), significantly decreased to nearly half of the non-tumors (Supplementary Data 5, Fig. 7d). Taken together, demethylation-induced high expression of Igf2/H19, followed by activation of PI3K/Akt and MAPK/Erk signaling might contribute to the tumorigenesis of Srsf2 HKO mice.

## Discussion

In this study, we showed that when severe liver damage impaired hepatocyte turnover following Srsf2 deletion, HPCs were induced to proliferate and expand. This was closely related with HCC

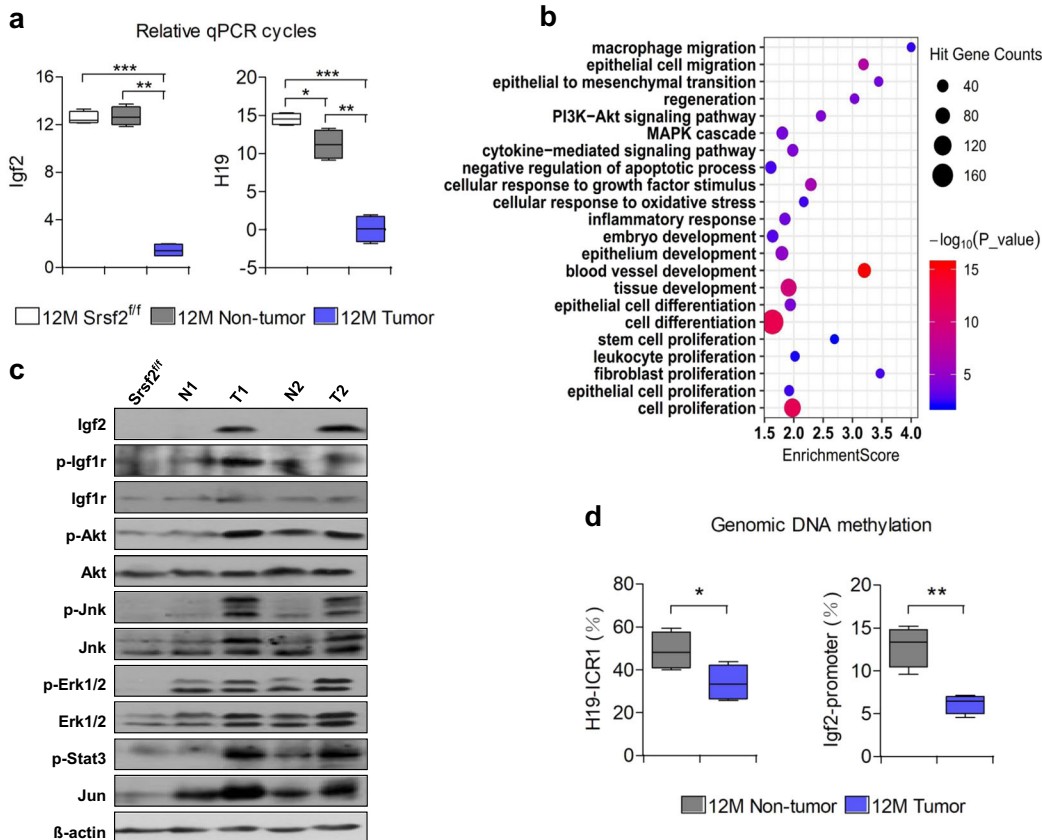

**Fig. 7 High levels of Igf2/H19 and activated pathways in the tumors. a** Comparison of Igf2 and H19 mRNA levels in 12M Srsf2$^{f/f}$, non-tumors or tumors of 12M mutant mice ($n = 3$). Note that qPCR cycles calculated as ΔCt values relative to the mean of β-actin. **c** Bubble diagram of potential pathways and biological processes based on Gene Ontology and KEGG analysis using the differentially expressed genes of 12M tumor versus non-tumors. **d** Igf2 and relevant pathways were activated in tumors of 12M mutant mice, addressed by western blotting analysis. Note that N for non-tumor and T for tumor; **e** Genomic CpG methylation ratios of H19 ICR1 and Igf2 P2 in 12M non-tumors and tumors ($n = 3$). *$P < 0.05$, **$P < 0.01$, and ***$P < 0.001$ vs. matched control by Student's $t$ test (**b, d**).

development in Srsf2 KO mice, as cells in HCC-bearing livers expressed both HPC markers and hepatocyte markers. The high levels of Igf2 and H19, together with multiple activated signaling pathways could further promoted HCC development.

Our results suggested that HPC activation, proliferation and expansion could be responsible for HCC development. First of all, inactivation of Srsf2 either in hepatocytes by Alb-Cre or in adult hepatocytes by Mx1-Cre triggered substantial P21 acceleration, which impaired hepatocyte replication that in turn could induce the activation of HPCs. Secondly, HPCs that were present in the parenchymal regions in HKO mice gained some features of hepatocytes. This was similar to the phenomena caused by the long-term DDC treatment[4]. Thirdly, tumors developed on the Srsf2 HKO mice were composed of cells bearing both HPC markers (A6 and Sox9) and hepatic markers (Hnf4α). Notably, gene expression profiling by RNA-seq analysis further revealed that tumor cells expressed both HPC markers and oncofetal markers, and similar gene signatures were also observed in HCC induced by hepatic deletion of *PR-SET7* in mice and in human aggressive HCC[9,10]. These findings strongly suggested that HPC expansion and possible transformation made tremendous contributions to HCC development in the Srsf2 HKO livers, even though we did not completely exclude that hepatocytes could also derive from pre-existing hepatocytes.

Human HCC comprises several distinct biological phenotypes with different prognosis, which may indicate a different cell of origin or activation of different oncogenic pathways during HCC development. In mouse models, inactivation of many genes has been

reported to trigger HCC formation and development[10,20,32,33]. The common features among them were that liver cell damage, inflammation and fibrosis persisted in the early stages of HCC, which occurred during the onset of human tumorigenesis. In addition, no tumors were detected in the Srsf2$^{-/-}$ mice following 11 months of polyI:C treatment (Supplementary Fig. 10), suggesting the requirement of chronic and extensive injury during the tumor development. Lineage tracing experiments suggested that HCC originates from hepatocytes instead from the biliary compartment[11], and benign lesions of liver could originate from HPCs[12]. However, tumors induced by hepatic deletion of *Srsf2* or *PR-SET7* in mice suggested that HCC might result from HPCs, which seemed to be similar to that observed in human aggressive HCC.

The gene expression analysis showed that several embryonic genes were induced in the tumors of HKO mice, among which Igf2 and H19 were strongly up-regulated. IGF2 was up-regulated in hepatitis B virus-related HCC, and IGF2 methylation decreased during progression from cirrhosis to HCC[34,35]. Elevated levels of IGF2 was also observed in a large proportion of HCC through demethylation of its fetal promoter, which could accelerate formation of liver tumors in mice via activation of IGF1R signaling[36]. Based on research, IGF2 has been proposed as a potential driver in hepatocarcinogenesis, but the evidence is still missing. Therefore, it would be informative to test whether inhibition of Igf2 signaling could suppress tumor formation in the Srsf2 HKO mice model in the future.

In summary, we provided strong evidence that Srsf2 deletion triggered HPC-mediated regeneration and activation of Igf2

signaling pathway. These processes could be closely linked to the development of HCC in mice.

## Methods

**Animals and experiments.** Srsf2[f/f] mice (kindly provided by Dr. Fu XD) were crossed with Alb-Cre mice to generate Srsf2 HKO mice as described previously[13,19]. Srsf2[f/f] mice were crossed with Mx1-Cre mice expressing Cre under the control of type I IFN-inducible Mx1 promotor[37] to generate Srsf2[f/f]-Mx1cre mice. 10 to 12-week-old Srsf2[f/f]-Mx1cre mice were intraperitoneal injection with polyinosinic:polycytidylic acid (polyI:C, InvivoGen) to produce Srsf2[−/−] mice. Srsf2[f/f] mice were received equal amounts of polyI:C as controls.

To induce the chronic liver injury, Srsf2[f/f] mice were fed with a standard chow diet containing 0.12% wt/wt 3,5-diethoxycarbonyl-1,4-dihydro-collidin (DDC) (Sigma) for 6 weeks or more. All experiments were conducted in accordance with the guidelines of the Institutional Animal Care and Use Committee of the Institute for Nutritional Sciences, Shanghai Institute for Biological Sciences, Chinese Academy of Sciences.

**Blood chemistry.** ALT and AST were measured using commercial kits (Shanghai Shensuo UNF Medical Diagnostic Articles Co., Ltd.) following the manufacturer's instructions.

**RNA extraction, cDNA synthesis, qPCR and western blotting analysis.** RNA extraction, cDNA synthesis, qPCR and western blotting analysis were conducted as previously described[38,39]. Briefly, total RNA was extracted from livers using the Total RNA Isolation Reagent (Pufei, China, 3102-100). cDNA synthesis was performed using oligo (dT) priming and M-MLV Reverse Transcriptase according to the manufacturer's instructions (Promega Corporation, Madison, WI). For qPCR analysis, quantification of all gene transcripts was carried out by real-time PCR using the TB Green[TM] Premix ExTaq[TM] kit (Takara, RR420A). β-Actin was used as control. For evaluation of Igf2 and H19 mRNA levels, their relative qPCR cycles were calculated as ΔCt values relative to the mean of β-actin. For detection of Srsf2 proteins in liver tissues, liver nuclear proteins were extracted using a commercial kit (KeyGen Biotech) following manufacturer's instructions. Antibody lists and the sequences of primer sets were shown in Supplementary Data 2.

**Histochemistry, immunostaining and TUNEL assay.** Collected livers were fixed with 4% paraformaldehyde for 4 h and dehydrated in gradient ethanol or 30% sucrose overnight, which were then embedded with paraffin or 22-oxacalcitriol, respectively. Paraffin sections were used for staining of HE, PAS, TUNEL assay, and immunohistochemical assay via detection of oxidized 3,30-diaminobenzidine substrate while frozen sections were used for staining of oil-red were performed as previously described[18,19]. Immunofluorescence on the paraffin tissues were carried out by standard procedure. Briefly, 5μm paraffin tissues were deparaffinized in xylol and rehydrated with gradient ethanol followed by antigenic retrieval through 98 °C water. Then the sections were treated with 0.3% hydrogen peroxide to eliminate the endogenous peroxidase activity followed by incubation with 1% and 5% goat serum, respectively. Immunostaining on frozen sections was conducted as previously described[40].

**EdU assay.** In vivo proliferation rate was determined using a Click-iT EdU Cell Proliferation Kit (C10337; Invitrogen). In brief, 1M Srsf2[f/f] or HKO mice were intraperitoneally injected with EdU (3.5 μg/g body weight) and sacrificed 5 days later. EdU was detected using the Click-iT reaction cocktail according to the manufacturer's instructions.

**RNA-seq data and GO/KEGG enrichment analysis.** Total RNAs were extracted from livers of 2M Srsf2[f/f], 2M HKO, 12M Srsf2[f/f], 12M non-tumorous section or 12M tumorous section of mutant mice, which were used for cDNA library construction and paired-end reads sequencing. Statistical tests in differential gene expression analysis of corresponding pairs (i.e. 2M HKO versus 2M Srsf2[f/f], 12M non-tumorous section versus 12M Srsf2[f/f], and 12M tumorous section versus non-tumorous section) were performed by EBSeq R package[41]. Differentially expressed genes (DEGs) were selected with p value < 0.05, log₂FC > 1 or < −1, and FPKM > 5 in at least one condition. Volcano plot showing dysregulated genes between the 12M tumors and non-tumors was generated using R programming. GO and KEGG enrichment analysis were performed on the DEGs using TBtools.

**Comparison of our RNA-seq data with published microarray and RNA-seq data.** Based on microarray data of adult liver progenitor-enriched cells and depleted cells[26] from NCBI GEO (GSE29121), we obtained differentially expressed probes using GEO2R tool[42] and selected the probe with the lowest p-value to represent the corresponding gene. Subsequently, we compared the DEGs (p value < 0.05 and log₂FC > 1 or < −1) from the microarray data[26] with our RNA-seq DEGs to obtain overlapping up- and down-regulated genes.

**Genomic DNA isolation and methylation analysis.** Genomic DNA was isolated from tumorous and non-tumorous liver sections of Srsf2[f/f] or HKO mice using a commercial kit following manufactures' instructions. To quantitatively assess DNA methylation levels in ICR of H19 or in fetal promoters of Igf2, we adopted bisulfite sequencing PCR (BSP) method as previously described[43]. In brief, 2 μg of genomic DNA was bisulfite converted using the EpiTect Bisulfite Kit (Qiagen), and target regions were amplified by PCR and cloned to the pMD[TM]18-T vector. Ten clones were randomly selected from each sample for sequencing, and the ratio of methylated CpG sites to total CpG sites was calculated.

**Statistics and reproducibility.** Data are expressed as mean ± SD of the total number of independent experiments. Non-numerical data are shown as representative results of three independent experiments. Statistical analysis was performed by Student's t test at a significance level of *P < 0.05, **P < 0.01, ***P < 0.001.

**Reporting summary.** Further information on research design is available in the Nature Research Reporting Summary linked to this article.

## Data availability

The RNA-seq data have been deposited to Gene Expression Omnibus (GEO) with accession number GSE130745.

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

## Acknowledgements

This work was supported by grants from the Shanghai Scientific Research Project (19JC1416000), the National Natural Science Foundation (31870819, 31570818), and also from the "Personalized Medicine-Molecular Signature-based Drug Discovery and Development", Strategic Priority Research Program of the Chinese Academy of Sciences (XDA12010100).

## Author contributions

C.Z. designed and performed the experimental work. L.S., W.Y., and G.W. performed pathological analysis. Yu.L., Ya. L, R.G., Z.Z., and Z.X. provided technical support and animal experiment. W.W. helped with RNA-seq analysis and other data analysis. C.Z. and Y.F. performed data analysis and wrote the manuscript. All authors reviewed and edited the manuscript.

## Competing interests

The authors declare no competing interests.
