## [Peer Review File · Communications Biology]

Reviewers' comments:

Reviewer #1 (Remarks to the Author):

This is an interesting manuscript which identifies that deletion of *Srsf2* within the results in development of hepatocellular carcinoma (HCC). The experiments shown are well performed but there are a number of mechanistic points which should be clarified to improve the content and clarity of the manuscript as follows:

-SRSF2 is an RNA binding protein and splicing regulator. There are no data presented on the effects of *Srsf2* deletion on RNA splicing and there are no links presented between RNA splicing and the gene expression targets described here (such as *Fgf7*, *H19*, and *Igf2*). Are any of the target genes described altered in RNA splicing directly by *Srsf2*?

-What is the mechanistic basis for altered DNA methylation in *Igf2* locus here and how is this related to deletion of *Srsf2*?

-Is *Srsf2* deleted in the eventual HCC tumors that develop in the Alb-Cre *Srsf2* floxed mice? This is important to shown given the extent of time required for these tumors to develop and the apparent requirement of cell non-autonomous inflammation on HCC development in this model.

-As a minor point related to the above, the authors should describe the exact genetically engineered mouse model being studied at initial discussion of this model in the RESULTS section (the fact that Alb-Cre was used is described later in the Results/Discussion and Methods but this needs to be stated at the first use of this model in the Results section for clarity). Also, the authors should describe at which point in liver development and in what exact hepatic cell types Alb-Cre influences (this can come from the literature on this model). This point is highly relevant since the authors describe effects of *Srsf2* deletion on hepatocyte progenitors as well as hepatocytes.

-Do the Mx1-cre *Srsf2* floxed mice develop HCC? This point is not clear in the manuscript and is very relevant as *Srsf2* would also be deleted in bone marrow derived hematopoietic cells in this model and this could influence inflammation within the liver.

-The final paragraph of the Discussion describes "Srsf1" but I believe the authors meant to refer to *Srsf2*.

Reviewer #2 (Remarks to the Author):

The manuscript by Zhang et al investigates the loss of SRSF2 in the adult liver and seeks to define how and why loss of SRSF2 causes oval cell proliferation and cancer formation.

Generally the paper is well written, however it needs a thorough proof read as there are multiple obvious typos throughout and these need to be caught and corrected, for example, the use of "mice could liver beyond..." and "SRSR2". The authors also use oval cells and HPCs interchangeably. You should chose a constant nomenclature.

The manuscript could also benefit from being more specific - give absolute values of how many mice developed HCC (not simply saying upto 80%). or approximately 30% of HKO mice...you have done the experiments, give the data.

Please have a pathologist diagnose the tumours - are they HCC or not? Some of the makers you have used are associated with HCC, but this is not the same as a pathological diagnosis.

I have a query about the repeated assertion that this is spontaneous cancer - the deletion of SRSF2 gives underlying disease, fibrosis and repair. This is not spontaneous.

Have the authors looked what happens when SRSF2 is deleted in a mosaic or focal way? is local loss of SRSF2 sufficient for cells to form cancer - or is chronic and extensive injury required?

I am surprised that the deletion of SRSF2 alone is sufficient to drive these phenotypes, however as the authors show, the markers of injury and proliferation are reduced after 3 months. Have the authors looked at the upregulation of other SRSF transcripts and proteins to see whether these is compensation?

As the authors suggest that there is an increased proliferation of hepatocytes following SRSF2 deletion - can they formally show this, either functionally by demonstrating that mice with the deletion regenerate their liver more quickly following a partial hepatectomy or by serially injecting with BrdU/FdU/EdU to look at cell cycle length to see whether they are actually different?

The levels of SRSF2 in Figure 4B are not impressively reduce (although I note that the housekeeping gene is also higher in these). It would be useful to show with a standard loading amount of protein the level such that we can evaluate whether the SRSF2 protein is actually lost completely or not.

Throughout Figure 4, comparisons between the DDC model and SRSF2 deletion are interesting. Could these be formally quantified and presented as numerical data in addition to the images.

I find the section on HPC transformation (pg7) very weak and the link between HPC transformation and hepatocyte differentiation in this manuscript is not compelling - you should tone this down or show that the HPCs are becoming hepatocytes by lineage tracing.

Similarly, on pg9 using the recurrence of SRSF2 expression as a proxy for HPC to hepatocyte differentiation is not sufficient - there is large amount of literature showing that rare hepatocytes can proliferate and repopulate the liver if they have a growth or selection advantage. It is perfectly likely that this is the case here unless you can show that 100% of hepatocytes have lost SRSF2 (which from your WB in 4b, doesn't seem to be the case).

Finally, the RNAseq data is interesting. Have the authors tried inhibiting IGF signalling to see whether it supresses tumour formation?

Our detailed response to the reviewers' comments is as follows (reviewers' comments in italics):

Reviewer #1 (Remarks to the Author):

This is an interesting manuscript which identifies that deletion of Srsf2 within the results in development of hepatocellular carcinoma (HCC). The experiments shown are well performed but there are a number of mechanistic points which should be clarified to improve the content and clarity of the manuscript as follows:

-SRSF2 is an RNA binding protein and splicing regulator. There are no data presented on the effects of Srsf2 deletion on RNA splicing and there are no links presented between RNA splicing and the gene expression targets described here (such as Fgf7, H19, and Igf2). Are any of the target genes described altered in RNA splicing directly by Srsf2?

We agree with the reviewer that Srsf2 is an important splicing regulator. In fact, we have shown in the previous publication (Mol Cell Biol, 2016) that deletion of Srsf2 could result in splicing alteration in a large set of genes in the livers of Srsf2 knockout mice at the postnatal day 11. But these altered splicing events were not detected in livers from Srsf2 knockout mice at 2, 6 or 12 months of age, which is consistent with the fact that Srsf2 proteins recurred in the knockout livers beginning at 2 months (Fig. 6b). In addition, we found that Srsf2 was not involved in splicing regulation of *Fgf7*, *H19*, and *Igf2* pre-mRNAs. Up-regulation of these genes were closely related to liver regeneration present in the Srsf2 knockout mice. We have added new data into Supplementary Fig. 3 and Supplementary Fig. 5, and modified the text accordingly (p9-10).

-What is the mechanistic basis for altered DNA methylation in Igf2 locus here and how is this related to deletion of Srsf2?

Up-regulation of *Igf2* and/or *H19* commonly occurs in liver regeneration and HCC induced by gene knockout in mice (Sen et al. *Hepatology*, 2010; Nikolaou et al. *EMBO*, 2015; Wang et al. *Cell Death & Disease*, 2018). This can be partially explained by the decreased methylation at the fetal promoters of *Igf2*. However, the mechanism for the loss of imprinting in *Igf2* gene still remain elusive.

In our study, many oncofetal genes were observed to be up regulated in the tumors developed on the knockout mice, among which the most obvious change was for *Igf2* and *H19* genes. To study whether high expression of *Igf2* and *H19* was closely related with Srsf2 loss, we chose 3 time points to analyze their expression levels: At 2 weeks after birth when knockout livers went through massive cell death due to loss of Srsf2 proteins, mRNA levels of *Igf2* and *H19* were significantly decreased in the knockout livers compared to controls. At 1 month of age when knockout livers had compensatory hepatocyte proliferation but still no expression of

Srsf2 proteins, mRNA levels of Igf2 or H19 were observed to be up-regulated more than 20-fold or 10-fold, respectively. At 2 months old when knockout livers had progenitor cell proliferation, stellate cell activation and recurrence of Srsf2 proteins, mRNA levels increased hundreds of times for Igf2, or thousands of times for H19. Thus, up-regulation of Igf2 and H19 was closely related with liver regeneration in the Srsf2 knockout mice model, which is consistent with previous literatures. And the mechanistic basis for altered DNA methylation in Igf2 and how is this related to Srsf2-induced liver regeneration would be our future focus. We have added new data into Supplementary Fig. 5 and modified the text accordingly (p10).

-Is Srsf2 deleted in the eventual HCC tumors that develop in the Alb-Cre Srsf2 floxed mice? This is important to shown given the extent of time required for these tumors to develop and the apparent requirement of cell non-autonomous inflammation on HCC development in this model.

Following the reviewer's suggestion, we isolated genomic DNA from livers of control or Srsf2 knockout mice and performed PCR analysis. Indeed, deletion of Srsf2 gene was detected in the tumors that develop in the knockout mice, as well as in livers of knockout mice at different ages. Recurrence of Srsf2 proteins in adult knockout mice can be attributed to cells in the livers that are not targeted by the Alb-cre construct (Fig. 6b). We have incorporated new data into Supplementary Fig. 2 and modified the text accordingly (p9).

-As a minor point related to the above, the authors should describe the exact genetically engineered mouse model being studied at initial discussion of this model in the RESULTS section (the fact that Alb-Cre was used is described later in the Results/Discussion and Methods but this needs to be stated at the first use of this model in the Results section for clarity). Also, the authors should describe at which point in liver development and in what exact hepatic cell types Alb-Cre influences (this can come from the literature on this model). This point is highly relevant since the authors describe effects of Srsf2 deletion on hepatocyte progenitors as well as hepatocytes.

Following the reviewer's suggestion, we have now modified the last paragraph of Introduction section (p3), in an effort to provide more details about the generation of Srsf2 knockout mice. In fact, the endogenous Alb gene is expressed exclusively in hepatocytes, and Alb mRNA is mainly expressed after birth (Supplementary Fig. 1a).

-Do the Mx1-cre Srsf2 floxed mice develop HCC? This point is not clear in the manuscript and is very relevant as Srsf2 would also be deleted in bone marrow derived hematopoietic cells in this model and this could influence inflammation within the liver.

Following the reviewer's suggestion, we have isolated livers from Mx1-cre Srsf2

floxed mice (*Srsf2*^{-/-}) at 11 months of age. But there were no tumors detected on the livers at this age. HE staining revealed increased anisokaryosis in the mutant livers. And fibrosis staining showed that fibrosis was much obvious in the *Srsf2*^{-/-} livers compared to the control. We have incorporated the new data into Supplementary Fig. 7 and modified the text accordingly (p12).

-The final paragraph of the Discussion describes "Srsf1" but I believe the authors meant to refer to Srsf2.

The review is correct, and we have changed it in the revised manuscript.

Reviewer #2 (Remarks to the Author):

The manuscript by Zhang et al investigates the loss of SRSF2 in the adult liver and seeks to define how and why loss of SRSF2 causes oval cell proliferation and cancer formation.

Generally the paper is well written, however it needs a thorough proof read as there are multiple obvious typos throughout and these need to be caught and corrected, for example, the use of "mice could liver beyond..." and "SRSR2". The authors also use oval cells and HPCs interchangeably. You should chose a constant nomenclature.

We appreciate these comments and minor corrections. Following the reviewer's suggestion, we have used "track changes" in a word processing in an effort to improve the manuscript. And we also asked our colleagues to review the manuscript to find typos and incorrect word choices.

The manuscript could also benefit from being more specific - give absolute values of how many mice developed HCC (not simply saying upto 80%). or approximately 30% of HKO mice...you have done the experiments, give the data.

Following the reviewer's suggestion, we have provided absolute values as follows. For example, among a total of 136 HKO mice, 40 mice survived into adulthood. And 18 mice have developed HCC from a total of 25 mice at ages 12M and over.

Please have a pathologist diagnose the tumours - are they HCC or not? Some of the makers you have used are associated with HCC, but this is not the same as a pathological diagnosis.

Following the reviewer's suggestion, we have asked a pathologist from Zhongshan hospital (Shanghai) for helping us diagnose the tumors based on the HE staining. We have added new data into new Fig. 1f, and modified the text accordingly (p4).

I have a query about the repeated assertion that this is sponaneous cancer - the

deletion of SRSF2 gives underlying disease, fibrosis and repair. This is not spontaneous.

We are sorry for repeated asserting “spontaneous tumors” developed in the HKO mice. Following the reviewer’s suggestion, we have deleted “spontaneous” in the title and in the text as well.

Have the authors looked what happens when SRSF2 is deleted in a mosaic or focal way? is local loss of SRSF2 sufficient for cells to form cancer - or is chronic and extensive injury required?

We have carefully examined livers of heterozygous *Srsf2* mice at 12 months of age, but no tumors were developed at this age. We also had Mx1-cre *Srsf2* floxed mice in which inactivation of *Srsf2* genes starts in adult hepatocytes. However, we have not observed liver tumors in the mice of 11 months after polyI:C treatment. Thus, it is very likely that chronic and extensive injury were required for tumor formation in the *Srsf2* knockout mice. We have added the new data into Supplementary Fig. 7 and modified the text accordingly (p4, p12).

I am surprised that the deletion of SRSF2 alone is sufficient to drive these phenotypes, however as the authors show, the markers of injury and proliferation are reduced after 3 months. Have the authors looked at the upregulation of other SRSF transcripts and proteins to see whether these is compensation?

Following the reviewer’s suggestion, we have analyzed the mRNA levels of other SR family genes by RT-qPCR using total RNA isolated from livers of *Srsf2*^{f/f} mice or HKO mice at 12 months old. However, there was no obvious changes in 8 transcripts examined between *Srsf2*^{f/f} mice and HKO mice. Fully in line with this, RNA-seq analysis of 12-month mice also revealed that there is no change in the expression of SR family members. We have added new data into Supplementary Fig. 4 and modified the text accordingly (p9).

In addition, we found that tumors could be induced by knockout of other genes in mice (Sen et al. *Hepatology*, 2010; Nikolaou et al. *EMBO*, 2015), which we mentioned in the Discussion section (p12).

As the authors suggest that there is an increased proliferation of hepatocytes following SRSF2 deletion - can they formally show this, either functionally by demonstrating that mice with the deletion regenerate their liver more quickly following a partial hepatectomy or by serially injecting with BrdU/FdU/EdU to look at cell cycle length to see whether they are actually different?

Following the reviewer’s suggestion, we injected HKO mice at 1 month of age with EdU, and analyzed liver sections using EdU assay after 5 days of injection. Fully in line with the results showed in Fig. 2d, we observed that Edu⁺/Hnf4a⁺ cells were

significantly increased in the liver section of HKO mice when compared with controls. We have added the new data into new Fig. 2e and modified the manuscript accordingly (p6).

The levels of SRSF2 in Figure 4B are not impressively reduce (although I note that the housekeeping gene is also higher in these). It would be useful to show with a standard loading mount of protein the level such that we can evaluate whether the SRSF2 protein is actually lost completely or not.

Following the reviewer's suggestion, we have adjusted protein content and performed western blotting analysis again using samples showed in Fig. 4b. As shown in the new Fig. 4b, Srsf2 protein was deleted completely in the livers harvested at 1 month or 2 months after injection with polyI:C in mice.

Throughout Figure 4, comparisons between the DDC model and SRSF2 deletion are interesting. Could these be formally quantified and presented as numerical data in addition to the images.

Following the reviewer's suggestion, we have quantified the image data and presented them in the form of histograms just below the images.

I find the section on HPC transformation (pg7) very weak and the link between HPC transformation and hepatocyte differentiation in this manuscript is not compelling - you should tone this down or show that the HPCs are becoming hepatocytes by lineage tracing.

We totally agree with the reviewer that only lineage tracing could tell the origin of cells.

We have toned down this paragraph in the revised manuscript.

Similarly, on pg9 using the recurrence of SRSF2 expression as a proxy for HPC to hepatocyte differentiation is not sufficient - there is large amount of literature showing that rare hepatocytes can proliferate and repopulate the liver if they have a growth or selection advantage. It is perfectly likely that this is the case here unless you can show that 100% of hepatocytes have lost SRSF2 (which from your WB in 4b, doesn't seem to be the case).

Now we realized that it is not appropriate using the recurrence of Srsf2 expression as a proxy for HPC to hepatocyte differentiation. We have modified this paragraph in the revised manuscript accordingly.

Finally, the RNAseq data is interesting. Have the authors tried inhibiting IGF signalling to see whether it supresses tumour formation?

A large amount of literature showed that IGF2 is highly overexpressed in hepatocellular carcinoma and other cancers (Martinez-Quetglas et al. *Gastroenterology*, 2016; Tovar et al. *J Hepatology*, 2010). IGF2 has been proposed as a potential driver in hepatocarcinogenesis, but the evidence is still missing. Therefore, it would be informative to test whether inhibition of IGF2 signaling could suppress tumor formation in the Srsf2 HKO mice model. However, it is very challenging and almost impossible to achieve enough HKO mice in a short period of time, for example, 10 mice in the experimental group and 10 mice in the control group. Because during the last 3 years, we have gotten 136 HKO mice, and only 40 of them could survive to adulthood. And 18 mice have developed HCC from a total of 25 HKO mice at ages 12M and over. Therefore, this will become our long-term goal in the future, which we have mentioned in the Discussion section (p13), and we appreciate the reviewer for this comment.

REVIEWERS' COMMENTS:

Reviewer #1 (Remarks to the Author):

The authors have addressed my initial questions and comments.

Reviewer #2 (Remarks to the Author):

I am content that my concerns have been met.